# Numerical Simulations of the NREL Phase VI Wind Turbine with Low-Amplitude Sinusoidal Pitch

Amir Akbarzadeh and Iman Borazjani *

J. Mike Walker Department of Mechanical Engineering, Texas A&M University, College Station, TX 77843, USA; amirmahd@tamu.edu
* Correspondence: iman@tamu.edu

**Abstract:** Currently, most wind turbine performance analyses and simulations are performed assuming constant pitch and yaw angles during each rotation. Nevertheless, induced vibration or rotor imbalance can affect the pitch or yaw angle within each rotation. In this study, the effects of low-amplitude sinusoidal pitch angle oscillations of the blade on the performance of a wind turbine was investigated numerically by comparing it against the baseline (without pitch variations). Large eddy simulations were performed in which the motion of blades was handled by the curvilinear immersed boundary (CURVIB) method. The grid resolution was increased near the moving immersed boundaries using dynamic overset grids to resolve rotating blades. It was found that low-amplitude (up to 3 degrees) sinusoidal oscillations in the pitch angle negligibly affected the mean torque but increased its fluctuations and created distinct features in the wake of the turbine. In fact, the turbine's mean torque at wind speed of 15 m/s decreases from 1245 N.m to 1223 N.m, while its fluctuation (standard deviation) increases from 2.85 N.m to 7.94 N.m, with a dynamic pitch of 0.5 degrees and frequency of 3.6 Hz.

**Keywords:** wind turbine; large eddy simulation; immersed boundary

## 1. Introduction

One of the fastest growing clean energies in the world is wind energy [1]. Consistent increases in wind energy require designing efficient turbine blades that maximize lift generation at each cross-section along the span of the blade. Most blades for wind power generation require optimized aerodynamic designs to maximize wind energy extraction. Indeed, the aerodynamic performance of a wind turbine depends on the aerodynamics and the design of its blade. To optimize the blade design, it is necessary to study the flow field using reliable experiments or simulations. However, these experiments are expensive due to the large size of wind turbines and issues associated with the measurements, e.g., they are often confounded by varying wind speeds, changes in wind direction, etc. Therefore, various computational fluid dynamics (CFD) studies with various levels of fidelity have been performed to investigate flow around the blade and full-scale wind turbines [2–4].

The main levels of fidelity in CFD include: (a) the blade-resolving approach, in which the flow over the blade is resolved; and (b) the blade element momentum (BEM) approach, such as actuator-disk [5], actuator-line [6], and actuator-surface models [7,8]. In the latter approach, the interaction of the rotor with the incoming flow is modeled by adding distributed forces on the rotor disk, along the blade lines, or on the blade surface. The BEM approach enables simulations of wind farms and their interaction with the atmospheric boundary layer, which is not possible even with current state-of-the-art supercomputers [7]. Nevertheless, such modeling approaches cannot be used to obtain the detailed flow over the rotor blades; for this reason, blade-resolving simulations have been performed [4,9–12]. The flow over the NREL Phase VI wind turbine, which has been measured comprehensively in the wind tunnel of NASA Ames [13], has served as a

benchmark case for many blade-resolving CFD simulations with different methods, such as the finite element [14], finite volume [15], and finite difference methods [16]. This test case will serve as the baseline for this study as well.

The blade's performance can be changed significantly by changing the pitch angle, wind turbulence, yaw, and rotational speed regulations [17]. Previous studies have shown that there is an optimum pitch angle for every wind speed [18,19]. Moreover, a pitch control can effectively reduce the fatigue load of turbines [20]. Santoni et al. [21] coupled large-eddy simulations with an actuator-surface model for wind turbines with control modules for active pitch control of each blade. Apart from active pitch variations, passive pitch variations may be induced by wind turbulence or rotor imbalance, which can affect the blade's aerodynamics and its performance. The effect of dynamic pitch oscillations has been investigated on airfoils numerically and experimentally. For example, Liu et al. [22] studied dynamic pitch of an S809 airfoil (the airfoil of the NREL phase VI turbine), and Kuches-fahani et al. [23] and Guillaud et al. [24] investigated dynamic pitching of NACA0012, reporting thrust generation (negative drag) at a small angle of attack. Based on these studies, therefore, the dynamic pitch of a blade might improve the performance of the wind turbine. Nevertheless, there is no prior work on dynamic pitch of a turbine blade in the literature, and the majority of studies on blade performance optimizations are focused on passive and active flow control [25]. In fact, the effect of the blade's vibration in the form of sinusoidal dynamic pitch has not been investigated before due to the complexity of the wind turbine's simulation and experiments. To explain the complexity, one should note that the angle of attack varies across the blade's span due to the different blades' relative velocities, i.e., the blade's velocity to the wind velocity and the blade's twist angle. On the other hand, the dynamic pitch changes the angle of attack constantly across the turbine's blade, which causes a variable range of angle of attack along the blade's span. Moreover, a dynamic pitch over a rotating blade generates a Coriolis force that might influence the wake and performance of the blade. To the best of our knowledge, these are the first simulations of a wind turbine that resolve a dynamically pitching blade.

In this study, a large eddy simulation (LES) of the NREL Phase VI wind turbine for two different wind speeds of $U = 7$ m/s and $U = 15$ m/s is studied. First, the framework is validated by comparing the results against experimental measurements with a fixed pitch angle. Next, the effect of a low-amplitude dynamic pitch on the performance of the NREL Phase VI wind turbine for both wind speeds is investigated. This is achieved by using large eddy simulations (LESs) through the recently developed dynamic overset-curvilinear immersed boundary (overset-CURVIB) method [26], which enables us to efficiently increase the grid resolution near the moving immersed boundaries (rotor blades) by curvilinear overset grids that rotate with the blades. The setup and the method are described in Sections 2 and 2.1, respectively. A grid sensitivity study is performed, and the simulations are validated against the NREL phase VI turbine measurement with constant pitch (Section 3.1). Afterwards, the results with vibrating blades with a sinusoidal pitch are presented in Section 3.2. Finally, the dynamically pitching blade is compared against the baseline, and conclusions are drawn (Section 4).

## 2. Materials and Methods

In this section, simulations of the NREL Phase VI wind turbine are performed using our dynamic overset framework [26]. NREL Phase VI is a modified Grumman Windstream turbine with full-span pitch control and a power rating of 18 kW. The turbine has the following three components shown in Figure 1c: (1) tower, (2) nacelle: a housing at the top of the wind turbine tower that contains the generator, gearbox, and other components, and (3) blades. The NREL Phase VI has two blades with an S809 tapered and twisted blade profile, as shown in Figure 1. The rotor diameter is $D = 2R = 10.058$ m, its height is 12.19 m, and the blade's maximum chord length is $L = 0.72$ m, which is at 0.25$R$, as shown in Figure 1. Three different types of simulations are performed. First, LES of flow over rotating blade was performed to validate the framework. Afterwards, simulations

of a full-scale turbine that includes the nacelle and tower are performed to investigate the effect of the tower and nacelle on flow. Finally, the simulations of rotating blades with a low-amplitude sinusoidal pitch are carried out. The turbine has an upwind configuration, the blade tip pitch angle is 3°, the yaw and cone angles are 0°, and the rotational speed is 72 rpm.

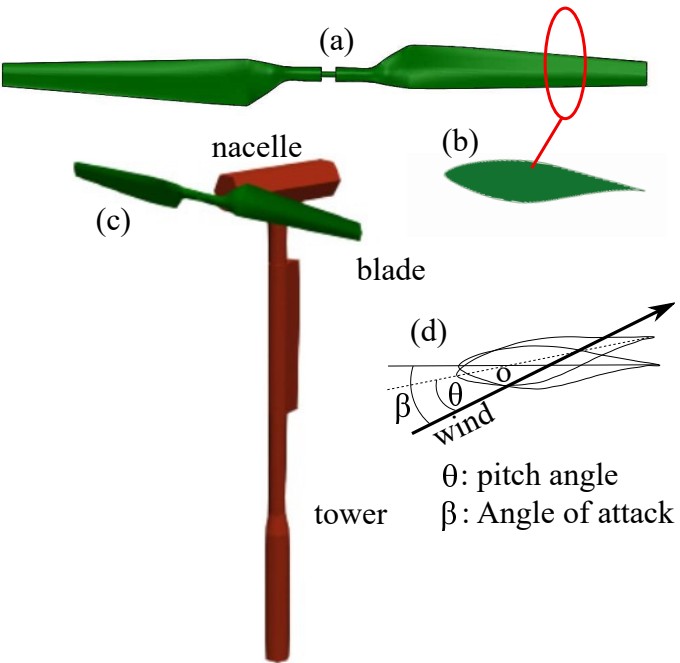

**Figure 1.** The wind turbine configuration: (**a**) blade, (**b**) its airfoil profile (S809), (**c**) complete turbine geometry, and (**d**) pitch angle definition.

The simulation setup is presented in Figure 2. The turbine has an upwind configuration, and the wind blows along the *z* direction. The setup has three cuboid grids where grid 1 is the background domain, grid 2 captures the turbine's wake, nacelle, and the tower, and grid 3 (overset grid) resolves the flow around the blade (Figure 2c). Grids 1, 2, and 3 have the size of $55L \times 55L \times 112L$, $22L \times 22L \times 24L$, and $16L \times 1.8L \times 2L$, respectively. Grids 1 and 2 are static, while grid 3 rotates with the rotating blade to increase the resolution near the blade. Because grid 3 should capture the blade's boundary layer, the validation and grid sensitivity studies are performed by changing the resolution of grid 3. Therefore, four sets of grid 3 are generated, while grids 1 and 2 are similar in all setups. The details of the grids' mesh sizes are presented in Table 1. The total number of meshes ranges from 30 million in setup 1 to 122 million in setup 4. The Reynolds number, based on the wind speed, the blade's maximum chord length ($L = 0.72$ m), and the kinematic viscosity of air ($\nu = 1.48 \times 10^{-5}$ m$^2$/s), is 345,000 and 739,000 for low and high wind speed cases, respectively. Moreover, the Reynolds number based on the blade's velocity (31.61 m/s), and the radius (5.03 m) is $1.27 \times 10^7$ at atmospheric condition.

**Table 1.** Setup of the simulations and their corresponding grids. Grids 1 and 2 are the same for all setups.

| Setup | Grid | $N_x$ | $N_y$ | $N_z$ | $\delta_{x_{min}}$ | $\delta_{y_{min}}$ | $\delta_{z_{min}}$ |
|---|---|---|---|---|---|---|---|
| 1 | 1 | 201 | 157 | 281 | $0.29L$ | $0.32L$ | $0.3L$ |
| 1 | 2 | 241 | 241 | 321 | $0.11L$ | $0.11L$ | $0.015L$ |
| 1 | 3 | 149 | 157 | 133 | $0.12L$ | $0.008L$ | $0.0057L$ |
| 2 | 3 | 301 | 273 | 201 | $0.06L$ | $0.004L$ | $0.0039L$ |
| 3 | 3 | 401 | 401 | 361 | $0.04L$ | $0.003L$ | $0.002L$ |
| 4 | 3 | 401 | 481 | 501 | $0.12L$ | $0.0026L$ | $0.0016L$ |

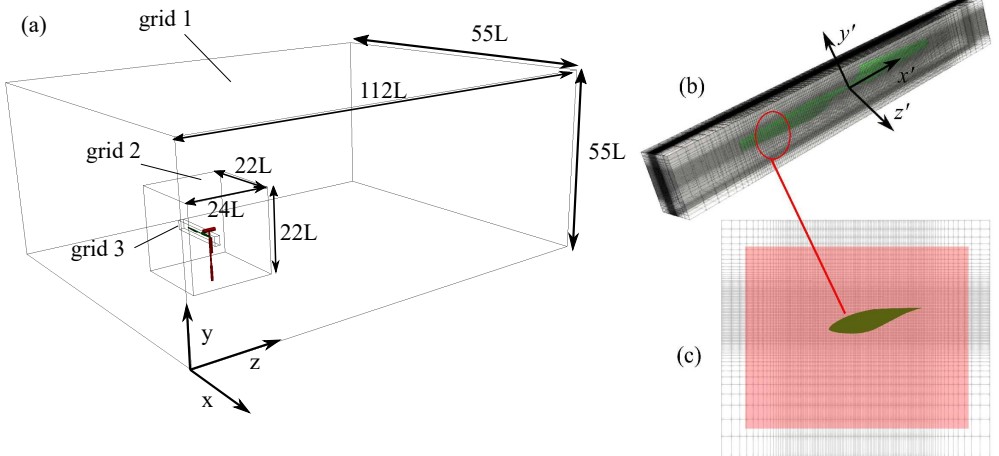

**Figure 2.** The wind turbine simulation setup. (**a**) The overset consists of three grids; grid 1 is the background domain, grid 2 includes the tower and nacelle, and grid 3 includes the rotor and rotates with the blades; (**b**) mesh grid of grid 3 and its frame, and (**c**) a cross-section of the blade (every four grid points along $y'$ and $z'$ are presented).

*2.1. Numerical Method*

Flow is considered to be incompressible, as the Mach number of the flow on the tip of the blade is 0.12. The flow in grids 1 and 2 is solved in an inertial frame, while grid 3 (moving overset grid) is solved in a non-inertial frame as it rotates with the blade. The equations governing the flow are the three-dimensional filtered unsteady incompressible continuity and the Navier–Stokes equations. The governing equations are first written in an inertial frame in the context of curvilinear coordinates ($\xi_i$: $i = 1$ to 3) and then transformed into non-inertial-frame curvilinear coordinates [26,27].

The filtered Navier–Stokes equations in curvilinear coordinates $\xi^i = \xi^i(x, y, z)$, are as follows, in tensor notation ($i, j, m, n = 1, 2, 3$):

$$J\frac{\partial U^i}{\partial \xi^i} = 0,$$

$$\frac{\partial U^i}{\partial t} = \xi^i_j\left(\frac{\partial}{\partial \xi^n}(-(U^n - V^n)u_j) - \frac{\partial}{\partial \xi^n}\left(\frac{1}{\rho}\frac{\xi^n_j}{J}p\right) + \frac{\partial}{\partial \xi^n}\left((\nu + \nu_t)\frac{g^{nm}}{J}\frac{\partial u_j}{\partial \xi^m}\right)\right).$$

(1)

Here, $\rho$ is the fluid density; $\nu$ is the kinematic viscosity; $J = |\partial(\xi^1, \xi^2, \xi^3)/\partial(x_1, x_2, x_3)|$ is the determinant of the Jacobian of the transformation $\xi^j_m = \partial\xi^j/\partial x_m$; $g^{nm}$ is the contravariant metric of the transformation, $g^{nm} = \xi^n_j\xi^m_j$; $p$ is the filtered pressure; $u_j$ and $v_j$ are the components of Cartersian fluid and grid velocities, respectively; $U^n = u_m\xi^n_m/J$ $V^n = v_m\xi^n_m/J$ are the contravariant velocity of the fluid and grid, respectively; and $\nu_t$ is

the subgrid-scale turbulent viscosity, which is modeled using the dynamic subgrid-scale model [28] as follows:

$$\nu_t = C_s \Delta^2 (|s|) s_{ij}, \tag{2}$$

where $s_{ij} = 0.5(\frac{\partial u_i}{\partial x_j} + \frac{\partial u_j}{\partial x_i})$, $|s| = \sqrt{2 s_{ij} s_{ij}}$, $\Delta = (J^{-1/3})$ is the filter size, and $C_s$ is the Smagorinsky constant computed by the curvilinear version [29] of the dynamic Smagorinsky model proposed by Lilly et al. [30] as follows

$$C_s = \frac{< L_{im} M_{jn} g^{mn} >}{< M_{im} M_{jn} g^{mn} >} \tag{3}$$

where

$$
\begin{aligned}
L_{ij} &= -\widetilde{u_i u_j} + \widetilde{u}_i \widetilde{u}_j \\
M_{ij} &= \widetilde{\Delta}^2 \widetilde{s}_{ij} \widetilde{|s|} - \Delta^2 \widetilde{s_{ij}|s|},
\end{aligned} \tag{4}
$$

where (~) represents the test filter, which is twice the grid filter ($\widetilde{\Delta} = 2\Delta$), and ($<>$) denotes averaging along the homogeneous direction, i.e., the test filter here, since there is no homogeneous direction. Several studies have shown the suitability of dynamic subgrid-scale modeling for turbulent and transitional flows [31]. Our LES modeling is also validated extensively for fully developed turbulent [32] and transitional flows, e.g., flows over pitching airfoils, inclined plates [33], and circular cylinders [34].

### 2.2. Overset-CURVIB Method

The moving boundaries are handled using the sharp curvilinear immersed boundary method (CURVIB) [35,36]. In this method, the background mesh is fixed, and the velocities of the fluid points adjacent to the moving boundaries (immersed nodes) are reconstructed using an interpolation along the surface normal [37]. The classification of the domain into solid, immersed, and fluid nodes is performed by a ray-tracing algorithm [36]. Here, the velocities of the immersed points are computed by applying the wall shear stress balance model proposed by Wang and Moin [38]. The details of development and validation of the wall model in CURVIB can be found in Akbarzadeh [39]. The overset-CURVIB is our in-house code developed in C and parallelized with MPI and PETSc [40].

The CURVIB method is validated for flows with moving boundaries [36,41] and has been applied in simulations involving turbulent flow [32,42] and in biological flows such as aquatic locomotion [34] and cardiovascular flows [43]. The overset method enables increasing the resolution locally near the immersed boundaries, thereby reducing the total number of grid points to achieve the same level of resolution compared to a single grid. The governing equations are in a non-inertial frame of reference for moving grids to avoid recalculating the curvilinear metrics of transformation, whereas the governing equations for stationary grids are in the inertial frame [26]. The boundary conditions on the overset boundaries are reconstructed by interpolation from the solution from nodes on other grids (called donor nodes) [26]. Due to the motion of the overset, the search needs to be performed at each time step, which is challenging when the interpolation and donor nodes are not on the same processor for parallel computations. Hedayat et al. [26] devised a parallel search and interpolation strategy for dynamically moving overset grids as encountered in wind turbines.

### 2.3. Computational Details

Using the overset framework [26], the velocities on the boundaries of grids 3 and 2 are interpolated from grids 2 and 1, respectively, via a second-order interpolation. Moreover, the velocities of the overlapping zones of grids 1 and 2 need to be interpolated from finer grids. To perform the interpolation of velocities for grids 1 and 2 from grids 2 and 3, respectively, some of the overlapping grid points are blanked out to transfer the information from the inner grid to the outer grid. For example, the pink area in Figure 2c represents the blank of grid 2. The velocities of the grid points over the blank are interpolated from

overlapped grid points of grid 3. The blank of grid 1 is a cube of $16L \times 1.5L \times 1.4L$ that has been cut by a cube of $2L \times 1.5L \times 0.4L$ from the surface that faces the nacelle and is centered with the blade's rotating center.

Herein, the flow of grid 3 is solved in a non-inertial frame, (Figure 2) while the flow in grid 2 is solved in the inertial frame. Therefore, after each interpolation, the interpolated velocity on the boundary of grid 3 is transformed to the non-inertial frame by a rotation. The interpolation of variables within the overlapped zone requires finding the suitable grid points in each iteration, since the blanking zone and grid 3 change due to rotation. The search for suitable grid points and interpolation of variables are performed using our parallel overset framework developed by Hedayat et al. [26].

The blade, tower, and nacelle are placed as immersed bodies, where the blade is in grid 3, the tower is in grid 2, and the nacelle is in both grids. The grid points are categorized into fluid, immersed, blank, and solid points by ray-tracing algorithms [36]. Here, the velocity of the immersed points is interpolated by Cabot wall modeling [13] because the boundary layer is turbulent and the grid normalized wall spacing along the normal direction is $z^{+} = u_{\tau}\delta_{z_{min}}/\nu = \mathcal{O}(500)$, where $u_{\tau}$ is the friction velocity.

The flow is initialized with a uniform velocity, $u_{z} = 1.0$. The inflow of grid 1 is a uniform (plug) flow, while the outlet of grid 1 is a Neumann boundary with a correction to satisfy the conservation of mass. The ground, which is along the $-y$ boundary, has a no-slip boundary condition on both grids 1 and 2. Each cycle of the turbine rotation has 8000 iterations for setup 1, 10,000 iterations for setup 2, and 20,000 iterations for setups 3 and 4. The CFL number for these cases ranges from 0.28 to 0.72, respectively. The simulations with setup 3 and 4 are performed for two cycles, although averaging was performed for one cycle due to high computational cost. Note that the aerodynamic torque becomes quasi-steady after about one-quarter of a cycle, and it has been observed that one-half of a cycle averaging is sufficient for reporting the mean values. Here, the averaging started after one cycle. It should be noted that high-fidelity overset simulations of wind turbines are computationally expensive. For example, the simulation cost of a baseline case at wind speed of 15 m/s is as follows. Those simulations are performed on the Terra cluster at the High-Performance Research Computing (HPRC) center of Texas A&M University with at least 16 nodes, each having 28 Intel Xeon E5-2680 cores, resulting in total 448 cores. One cycle of this calculation takes roughly 1220 h on 448 cores.

## 3. Results and Discussions

In this section, first, the results of the baseline simulations are presented in Section 3.1. To validate the simulations, the rotor torque and pressure coefficient are compared against the experiments with different setups. Afterwards, the effect of low-amplitude dynamic pitch is investigated in Section 3.2.

### 3.1. Baseline

Flow over the turbine's blade becomes separated at wind speeds greater than $U = 10$ m/s. One set of simulations is performed by only placing the rotor and another set by placing the entire turbine. Figure 3 shows the vortical structures identified by iso-surfaces of Q-criteria at $Q = 2$ for t the rotor simulations and the entire turbine simulations at wind speeds $U = 7$ m/s and $U = 15$ m/s. While strong tip vortices are generated in the wake of the low wind speed case, these tip vortices are not observed for high wind speed cases due to flow separation. Nevertheless, strong vortices can be observed along the nacelle and tower for the entire turbine simulation for both wind speeds (Figure 3c,d). Comparing the flow of the blade and the full turbine (Figure 3a,c) reveals the generation of strong vortices along the nacelle, as the nacelle does not have a streamlined shape.

The flow separation can be observed more clearly by visualizing the flow at different cross-sections. Figure 4 presents the contours of instantaneous out-of-plane vorticity and velocity vectors at three different radii, $r/R = 0.3$, $r/R = 0.80$, and $r/R = 0.95$. The velocity vectors represent the relative velocity of the air with respect to the blade. As shown in

Figure 4, the flow is attached for all cross-sections in the low wind speed case, but it is separated from the leading edge in the high wind speed case, as the angle of attack of the incoming air with the blade increases. In particular, the separation is higher at $r/R = 0.3$, since it has the highest angle of attack.

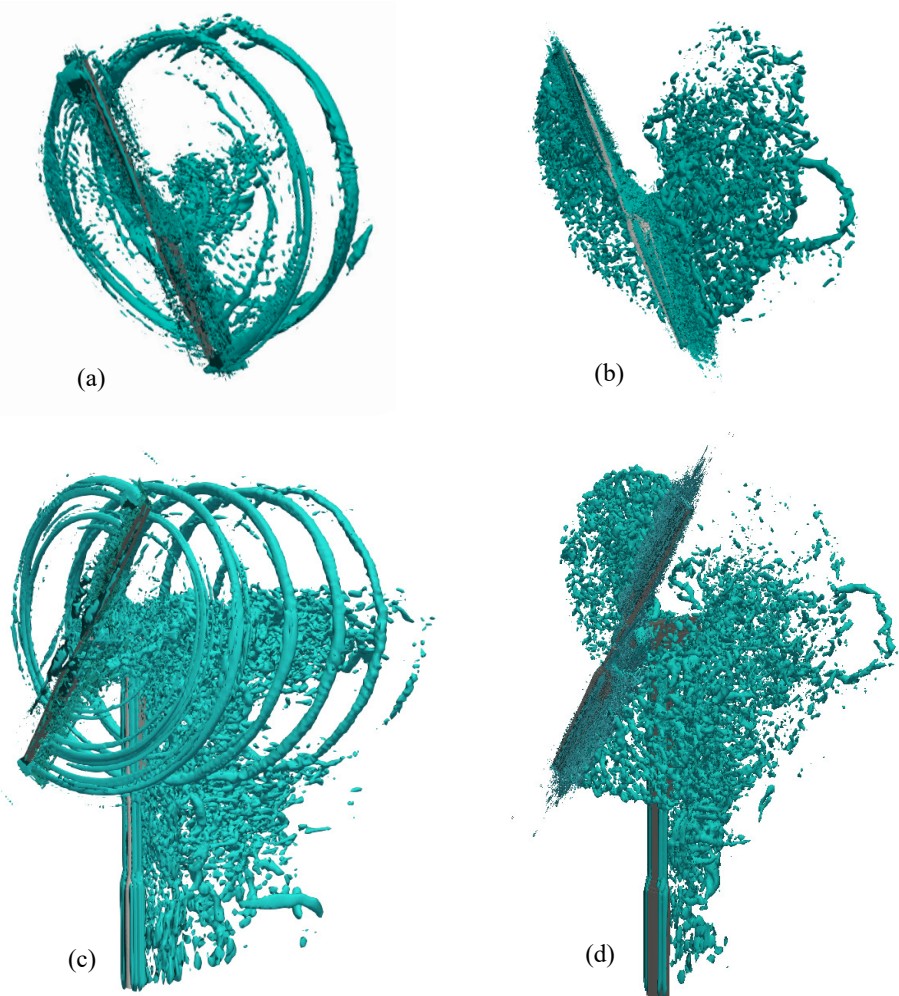

**Figure 3.** The wind turbine wake vortices identified by the iso-surface of Q-criteria with $Q = 2$ for the turbine's blade and the entire turbine. (**a**) Low wind speed ($U = 7$ m/s) blade, (**b**) high wind speed ($U = 15$ m/s) blade, (**c**) low wind speed ($U = 7$ m/s) entire turbine, and (**d**) high wind speed ($U = 15$ m/s) entire turbine.

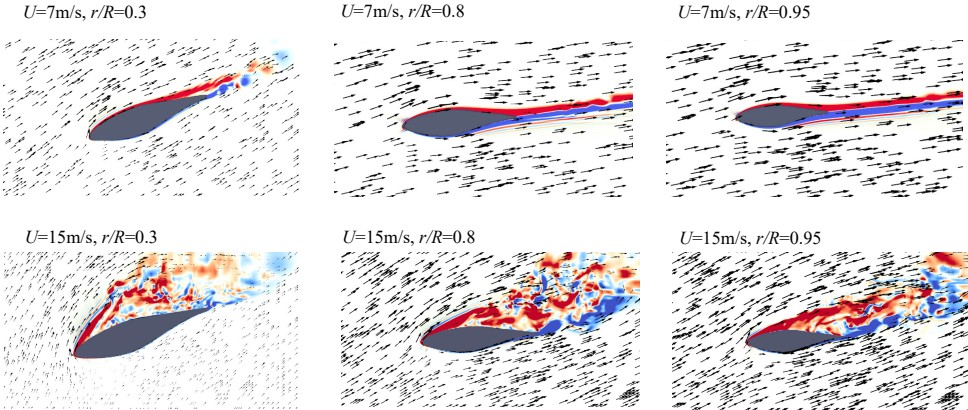

**Figure 4.** The contours of out-of-plane vorticity and velocity vectors.

Time-averaged aerodynamic torque is shown for both wind speeds in Figure 5. The rotor torque of the low wind speed case is plotted for all setups. The torque values are compared against NREL experiments [13]. The simulation results and experimental data match remarkably well at the low wind speed case for setups 2 and 3; however, the simulation results for setup 1 (coarsest grid) under-predicts the torque. Similarly, for the high wind speed case, the predicted torque for setups 3 and 4 is in good agreement with the experimental data, but setups 1 and 2 underestimate the torque due to lower grid resolution. It should be noted that the torque of the entire turbine is almost identical to the blade for the upwind configuration [44]. To extend the validations and grid sensitivity studies, the pressure coefficient over the blade is compared against the experimental measurements. The pressure coefficient is the pressure of the fluid points adjacent to the blade surface and is defined as follows:

$$C_p = \frac{p - p_0}{\frac{1}{2}\rho(U^2 + (r\omega)^2)} \tag{5}$$

where $r$ is the blade's radius, $\omega$ is the rotor speed, $U$ is the wind speed, and $p_0$ is the reference pressure, which is the pressure on the leading edge. The pressure coefficient is presented in Figure 6 for two setups, i.e., setups 2 and 3 for the low wind speed case and setups 3 and 4 for the high wind speed case at three blade radii, $r/R = 0.3$, $r/R = 0.8$, and $r/R = 0.95$. While $C_p$ for the low wind speed case agrees well with the experimental data on the pressure side for all radii, there is some discrepancy with the experiments on the suction side near the leading edge. For the higher wind speed case, $C_p$ matches with the experimental data on the pressure side; however, there is some discrepancy on the suction side for all radii near the leading edge, but it follows the trend well. The discrepancy with experiments at $r/R = 0.3$ is greater than other radii, which has been observed in previous numerical studies [14,15]. This discrepancy is due to the grid resolution of the Cartesian grid for capturing the separated boundary layer.

Nevertheless, the turbine's torque and $C_p$ plots (Figures 5 and 6) are in good agreement with the experiments, and it can also be observed that simulations are grid-independent when setups 2 and 3 are chosen for the low wind speed case and setups 3 and 4 are chosen for the high wind speed case.

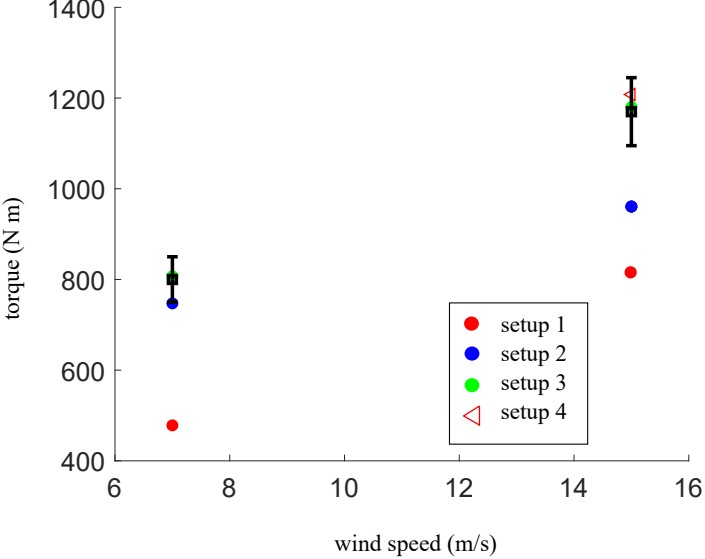

**Figure 5.** The wind turbine rotor torque for four simulation setups. The NREL data [13] are marked with black squares. The bars indicate the error of the measurements.

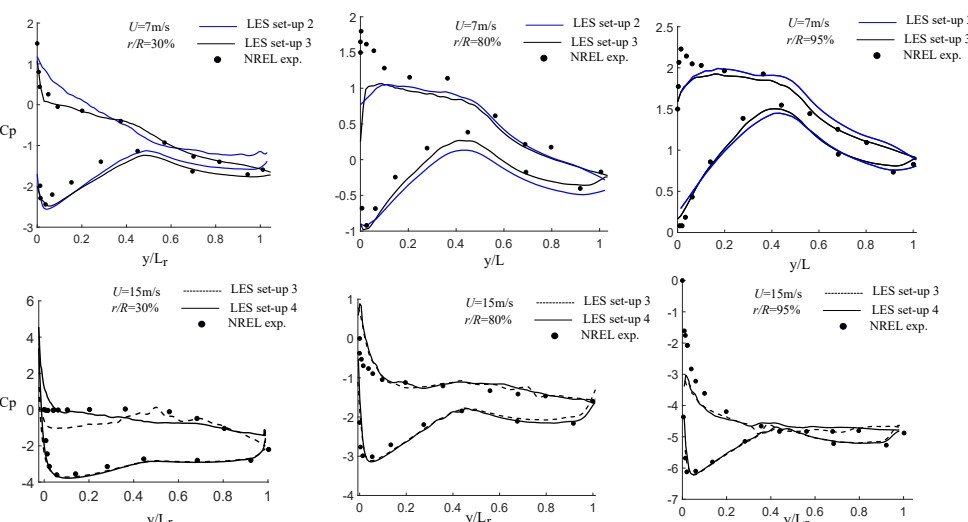

**Figure 6.** The pressure coefficient at 30%, 80%, and 95% of the blade radius.

### 3.2. Pitching Blade

In this part, the effect of low-amplitude sinusoidal dynamic pitch is investigated. The dynamic pitch is applied as a prescribed motion on the turbine's blade, and its equation is as follows:

$$\theta(t) = 3.0 + \theta_0 \sin(2\pi f t) \tag{6}$$

where $\theta$ is the blade's pitch angle, $\theta_0$ is the amplitude of the oscillation, and $f$ is the frequency of oscillations. Here, the frequency is 3.6 Hz, and $\theta_0$ is 3.0 and 0.5 degrees for low and high wind speed cases, respectively. This range of pitch angle is high enough to affect the turbine's blade performance. Previous studies have shown that a non-dynamic change of pitch angle by 3 degrees can change the turbine's power by 15% [19]. The amplitude of pitching for the high wind speed case is chosen to be small due to the computational cost and difficulty associated with high-amplitude oscillations in a refined mesh; therefore, the pitching blade simulations are performed with setups 2 and 3 for low and high wind speed cases, respectively. Moreover, the simulations are performed for rotating blades without the tower and nacelle to reduce the computational cost. The corresponding reduced frequency ($f^* = fL/U$, $L$: maximum chord length) of the pitching blade based on the wind speed and chord length is 0.36 and 0.024 for the low wind and high wind speed cases, respectively. On the other hand, the reduced frequency based on the blade's tip velocity ($U_{tip}$) $f_R^* = fL/U_{tip}$ is 0.08 and 0.011. The reduced frequency based on the wind speed and tip speed represent the ratio of the pitching velocity to the wind speed and blade's tip speed, respectively.

The mean torques for pitching cases at wind speeds $U = 7$ m/s and $U = 15$ m/s are 827 N.m and 1223 N.m, respectively, which are close to the baseline torques at the same speed (Figure 5), i.e., less than 1% change with respect to the baseline. The negligible change in the mean torque can be due to the small pitching velocity with respect the the blade's tip velocity, i.e., low $f_R^*$. In fact, for 2D wings and airfoils, a low reduced frequency, e.g., range of $f_R^*$, does not change the forces of a 2D pitching airfoil [23,32].

The instantaneous aerodynamic torque is plotted for both wind speeds in Figure 7. It can be observed that the dynamic pitch increases the torque fluctuations for both wind speeds. For example, the torque's fluctuation quantified by its standard deviation increases from 2.85 N.m to 7.94 N.m, with a dynamic pitch of 0.5 degrees and frequency of 3.6 Hz at wind speed of 15 m/s. In fact, the results of the simulations suggest that these low-amplitude dynamic pitches do not improve the mean aerodynamics and also increase the shaft oscillations, which can generate noise and damage the rotor.

Although the dynamic pitch tested here did not influence the mean torque, higher amplitude/frequency oscillations, e.g., higher $f_R^*$, might affect the mean torque, which needs to be tested in future work. The pitching motion can modify the tip vortex dynamics. Figure 8

compares the wake structure of the rotating blade with a dynamic pitch for both wind speeds. At the low wind speed, the tip vortex is still generated, but it has been deformed due to the blade's pitching motion. However, since the pitch angle was small, the flow did not separate, and strong tip vortices are present (Figure 8). On the other hand, at a high wind speed, the tip vortex of both baseline and pitching looks similar, as no tip vortex is generated. Moreover, the pitching motion can influence the turbulence statistics of the turbine's wake; however, investigating the wake's turbulence statistics is beyond the scope of this work.

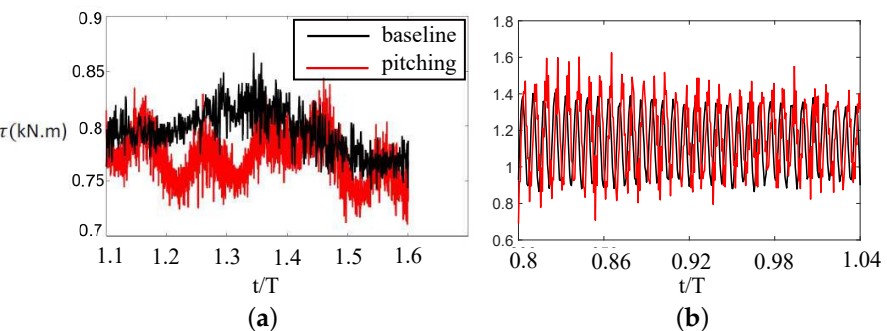

**Figure 7.** The instantaneous torque coefficient for the baseline and the pitching blade for (**a**) $U = 7$ m/s and (**b**) $U = 15$ m/s. Here, $T = 0.867$ s is a turbine cycle.

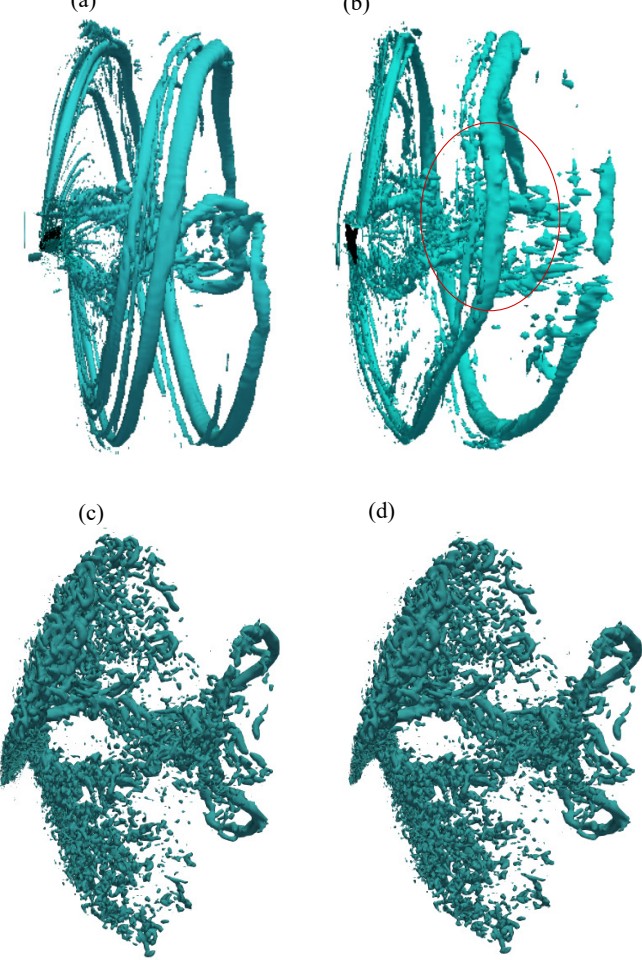

**Figure 8.** 3D vortical structures visualized by the iso-surface of Q-criteria for a blade at the wind speed of $U = 7$ m/s for (**a**) the baseline and (**b**) the pitching blade, and at wind speed of $U = 15$ m/s for (**c**) the baseline and (**d**) the pitching blade. The red circle shows the deformation of the tip vortex due to the pitching.

## 4. Conclusions

Large eddy simulations of a full-scale wind turbine were performed using our sharp-interface immersed boundary framework. The simulations were carried out for wind speeds $U = 7$ m/s and $U = 15$ m/s. The simulation mesh consists of 3 cuboid grids that contain the background, wake, and blade. The results of the simulations were validated against the experimental measurements by comparing the mean torque and pressure coefficients, and good agreement with the experiments was observed. The effect of low-amplitude sinusoidal dynamic pitch on the torque and flow was investigated. For both cases, the pitching did not change the mean torque but increased the torque fluctuations. For example, the turbine's mean torque at a wind speed of 15 m/s decreased from 1245 N.m to 1223 N.m, while its fluctuation (standard deviation) increased from 2.85 N.m to 7.94 N.m. Moreover, it was observed that the dynamic pitch can deform the tip vortex of the low wind speed case. The effects of high amplitude pitch and other types of vibrations (e.g., heave) and frequencies will be investigated as part of our future work.

**Author Contributions:** Conceptualization, I.B.; Methodology, A.A. and I.B.; Validation, A.A.; Formal analysis, A.A.; Investigation, A.A.; Resources, I.B.; Writing—original draft, A.A.; Writing—review & editing, I.B.; Visualization, A.A.; Supervision, I.B.; Project administration, I.B.; Funding acquisition, I.B. All authors have read and agreed to the published version of the manuscript.

**Funding:** This work is supported by National Science Foundation grant number CBET-1829408.

**Data Availability Statement:** The data that support the findings of this study are available from the authors upon reasonable request.

**Acknowledgments:** Computational resources were supported by the High-Performance Research Computing center of Texas A& M University.

**Conflicts of Interest:** The authors declare no conflict of interest.

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
