# Peer review of "Numerical Simulations of the NREL Phase VI Wind Turbine with Low-Amplitude Sinusoidal Pitch"

_fluids, doi:10.3390/fluids8070201_

Round 1
Reviewer 1 Report
Review on “Effect of low amplitude sinusoidal pitch on the performance and wake of NREL PHASE VI wind turbine”
by Akbarzadeh et al.
Manuscrip​t ID fluids-2414669
A- General Comments
The paper in hand concerns the effects of a low amplitude sinusoidal pitch angle oscillations of the blade on the performance of a wind turbine was investigated by comparing it against the baseline (without pitch variations). Large eddy simulations were performed in which the motion of blades were handled the curvilinear immersed boundary (CurvIB) method. Particularly, it was shown by authors that that low-amplitude (about 3 degrees) sinusoidal oscillations in the pitch angle negligibly affected the mean power output but increased its fluctuations, and created distinct features in the wake of the turbine.
The topic of the paper is interesting, within the scope of the journal, and worthy of investigation. The originality of the work is acceptable and the study performed is adequate. However, the manuscript deserves a major revision. I suggest that authors take into account the comments and questions below before it can be accepted for publication in fluids.
B- Detailed Comments and questions
Title
Details on how the study was performed can be helpful in the title. For instance, numerical study.
Abstract
1- More explicit results with numbers should be presented at the end of the abstract.
Keywords
Keywords are ok.
1- Introduction
1- References relevant to fluids should be added, if possible.
2- The literature review is insufficient and the originality of the work should be more highlighted at the end of the introduction especially with respect to the research gap in the field.
2- Material and Method
1- The title should read “Materials and Methods”.
2- Quality of figure 2 should be enhanced;
3- References to most of the equations presented should be provided.
3- Results
1- The title should read “Results and discussions”;
2- There are a lot of interesting observations without deep analysis. More physical analysis is to be added to this section by shortening the quantity of results shown if needed;
3- The quality of all figures of this section can be enhanced.
4- The close matching between numerical and experimental data should be argued further.
4- Conclusion
The main outputs of the work in terms of applications should be highlighted.
5- References
References relevant to fluids should be added, if possible.
The manuscript deserves an English proofreading.
Reviewer 2 Report
The manuscript entitled “Effect of low amplitude sinusoidal pitch on the performance and wake of NREL PHASE VI wind turbine” explores about wind turbine performance through pitch variation. The LES model is performed in the paper. The result suggested that low amplitude sinusoidal pitch negligibly affect the power output but increased the fluctuation. There is ample scope to improve the manuscript. Here are some revisions suggested to the authors:
1)What is the reason for considering the specific Curve IB method?
2) What is the deciding factor for time-step size?
3) How did the authors define Q criteria with Q=2? Did the value come from any formula or from CFD analysis?
4) Figure 7 represents torque with respect to a factor in the x-axis. What is that factor plotted on the x-axis?
5) Why did the authors select the amplitude of oscillation as 3 and 0.5 degrees? At what stage the authors implemented the blade’s pitch angle?
6) The following references may be cited in the manuscript:
https://doi.org/10.1051/e3sconf/202132103004
https://doi.org/10.1002/er.7488
Reviewer 3 Report
Kindly address the following
1. The abstract should incorporate numerical values pertaining to the turbine power coefficient Cp, considering the effect of low-amplitude (about 3 degrees) sinusoidal oscillations in the pitch angle.
2. The additional parenthesis in the abstract keywords (line 13) should be removed.
3. The passive voice should be used for lines 55 and line 59 (avoid using 'we' or 'enables us').
4. The unit of kinematic viscosity should be included in line 90.
5. Clearly specify in your text (line 89) the airflow velocity values corresponding to the two Reynolds numbers, Re=345000 and Re=739000.
6. How was the estimation of Re=10^7 made? Please provide a further explanation (line 92).
7. The dimensions of each component of your horizontal-axis wind turbine need to be included in Figures (a), (b), and (c) of Figure 1.
8. The dimensions of grids 1, 2, and 3 should be added to Figure 2.
9. Clearly specify the boundary conditions applied to your fluid domain in a diagram, for example, Figure 2-(a).
10. Do you use commercial software or a custom calculation code? Please specify in your article.
11. What are the specifications of your computational workstation (CPUs, RAM, launching method for calculations: serial or parallel)? Since LES calculations are computationally expensive, and it is challenging to run simulations with meshes ranging from 30 to 122 million elements, please provide these details in the revised version of your article to facilitate reproducibility by the scientific community.
12. Some information is missing regarding the estimation of the boundary layer around your blade. Please specify the value of y+ for the two velocities considered, as well as the distance y from the wall. This information is necessary for any CFD modeling around horizontal-axis wind turbine blades.
13. Did you use the sliding mesh technique or the multiple reference frame method? Please specify in your article, as you only mentioned inertial and non-inertial zones.
14. Please correct the caption of Figure 3. You forgot to include (c) and (d) in the caption.
15. When do you assume that the flow regime around your blade is established? After how many rotations of your machine? Please specify in the revised version of your article.
16. Please specify the CPU time for your LES modeling concerning the parametric study presented in subsection 3.2. Pitching blade.
17. The title of Figure 8 needs correction. You forgot to mention the two figures (c) and (d).
18. In the abstract, you mentioned the average power, but in the discussion subsection 3.2. Pitching blade, this quantity is not addressed. You either need to include it in your article or remove it from the abstract.
19. Please enhance your conclusion and abstract with the major numerical results obtained from your CFD modeling based on the LES turbulence model.
20. The "Results and Discussion" section must be enriched by incorporating a discussion that integrates recent references.
Moderate editing of the English language required
Round 2
Reviewer 1 Report
Thank you for taking into consideration my comments. The manuscript is now ready for publication.
English language is ok.
Reviewer 3 Report
The authors have addressed specific responses to all the comments I had proposed regarding the initially submitted version. It is for this reason that I recommend the publication of the current corrected version in its present form.